# Post-training Iterative Hierarchical Data Augmentation for Deep Networks

**Adil Khan***     **Khadija Fraz**
Institute of Data Science and Artificial Intelligence
Innopolis University
Universitetskaya St, 1, Innopolis, Russia, 420500
`a.khan@innopolis.ru, k.fraz@innopolis.university`

## Abstract

In this paper, we propose a new iterative hierarchical data augmentation (IHDA) method to fine-tune trained deep neural networks to improve their generalization performance. The IHDA is motivated by three key insights: (1) Deep networks (DNs) are good at learning multi-level representations from data. (2) Performing data augmentation (DA) in the learned feature spaces of DNs can significantly improve their performance. (3) Implementing DA in hard-to-learn regions of a feature space can effectively augment the dataset to improve generalization. Accordingly, the IHDA performs DA in a deep feature space, at level $l$, by transforming it into a distribution space and synthesizing new samples using the learned distributions for data points that lie in hard-to-classify regions, which is estimated by analyzing the neighborhood characteristics of each data point. The synthesized samples are used to fine-tune the parameters of the subsequent layers. The same procedure is then repeated for the feature space at level $l + 1$. To avoid overfitting, the concept of dropout probability is employed, which is gradually relaxed as the IHDA works towards high-level feature spaces. IHDA provided a state-of-the-art performance on CIFAR-10, CIFAR-100, and ImageNet for several DNs, and beat the performance of existing state-of-the-art DA approaches for the same networks on these datasets. Finally, to demonstrate its domain-agnostic properties, we show the significant improvements that IHDA provided for a deep neural network on a non-image wearable sensor-based activity recognition benchmark.

## 1 Introduction

Despite the tremendous success of deep neural networks in solving discriminative tasks, improving the generalization ability of these models remains as one of the most difficult challenges. There exist several ways to solve this problem, such as dropout, batch normalization, pretraining, transfer learning, and data augmentation (DA) [1, 2, 3, 4, 5]. The goal of DA is to present the predictor with a more comprehensive set of data points, during training, to minimize the distance between the training and the test sets.

DA increases the size of the training data set, generally, in two ways. The first approach expands the training data by performing expert-defined content preserving transformations to existing data points [2]. In the case of images, such alterations may include image rotation, flipping, random cropping, random erasing, and color space augmentation, etc. The second approach, on the other hand, inflates the training data through deep learning. These include feature space augmentation [6, 7, 8], adversarial training [9, 10], generative-adversarial-network based augmentation [11, 12, 13, 14], and meta-learning data augmentations (MLDA) [15, 16, 17, 18, 19]. MLDA employs prepended

---

neural networks to learn data augmentations; the learned augmentation strategies may include mixing images, neural style transfer, and a series of geometric transformations. Recently, [20] proposed a new DA method which performs implicit semantic data augmentation (ISDA). The ISDA is different from other DA methods because it does not require training/inferring auxiliary networks or explicitly generating extra training samples. It achieves DA implicitly by minimizing a derived closed-form upper bound of the expected cross-entropy loss.

There are some valuable lessons to be learned from the existing works on DA, and on machine learning, in general. The first is that DA is not as straightforward to apply in all domains as it is for images [8]. The second is that performing DA in the learned feature space instead of the input space can be beneficial, especially for domain-independent DA [7, 8]. The third is that the success of deep networks can be attributed to their ability to exploit the unknown structure in the input distribution to discover useful features at multiple levels. In this multi-level representations, the higher-level learned features are defined in terms of lower-level features [21]. Finally, the learning or classification difficulty associated with different samples in the feature space could be different [22].

In this paper, based on the lessons that we mentioned above, we propose and implement an iterative hierarchical data augmentation (IHDA) algorithm for deep networks. In IHDA, we do augmentation in the feature space, but unlike [7] which generates new samples in the feature space using extrapolation and interpolation, we generate new samples by learning a generative model over the feature space. Furthermore, unlike all existing works where the DA is performed once before or during the learning phase, we do it in the feature spaces of a learned model in an iterative manner, and use the augmented data to fine tune the model's parameters. More specifically, we start by training a deep network w.r.t a supervised loss on a given labeled training data without any augmented samples. Then, we generate augmented samples at each level of the representation learning part of the network and use them to fine-tune the subsequent levels of the representation learning part (results in learning of new hierarchical representations) as well as the predictor part of the network. Another vital property of the algorithm is that instead of blindly or randomly generating new samples for augmentation, it identifies regions that are hard to classify by assigning a *potential* value to each point in the learned feature space. Only the points having positive potential can act as sources for generating new data. Although it has a computationally expensive training phase, the proposed IHDA algorithm is significantly effective. Extensive empirical evaluations on several competitive image and non-image classification benchmarks showed that the IHDA consistently improved the generalization performance of popular deep networks and outperformed the existing state of the art DA algorithms for deep networks.

## 2 Related Work

Several methods have emerged over the past decade to reduce overfitting and improve the generalization performance of deep neural networks. These include dropout [23], batch normalization [24], transfer learning [25, 26], pretraining [27], few-shot learning [28], and DA. The focus of this work is DA, which can be classified and described in various ways.

**Explicit or Implicit:** DA can be explicit, such that it combats the overfitting problem by artificially increasing the size of training data through data warping or oversampling. Data warping-based DA generates new data by transforming existing data points while keeping their class labels preserved [1, 2, 3, 4, 5, 15, 16, 17, 18, 19]. The transformations could include elastic distortions, scaling, translation, rotation, mirroring, or color shift, etc. Oversampling-based DA works by expanding the training data through synthesizing new samples [11, 12, 13, 14, 22].

On the other hand, DA can be implicit [20] where the training procedure optimizes a surrogate loss function, instead of the original loss, to achieve augmentation without explicitly generating any extra training samples.

**Manual or Automated:** Explicit data warping-based DA can be a manual process where experts manually design dataset-specific augmentation strategies [1, 2, 3, 4, 5]. Or, it can be a completely automated approach where the augmentation policies are learned from the data using meta-learning techniques [15, 16, 17, 18, 19, 29].

Explicit oversampling-based DA either uses traditional methods like SMOTE (or its variants) [22] to create new data points or employ generative models like Variational Autoecoder or Generative Adversarial Networks for generating new data [11, 12, 13, 14].

**Input or Feature Space:** Finally, the DA can be implemented either directly in the input space [1, 2, 3, 4, 5, 15, 16, 17, 18, 19, 11, 12, 13, 14], or it can work by creating a feature space which is then manipulated to augment the training set [8, 30]. Terrance et al. [8] have demonstrated that because of the manifold unfolding in the feature space, by applying simple transformations to the encoded or learned representations of the data, more credible synthetic data can be generated than directly in the input space.

To summarize, the problem of DA has been extensively studied in literature. The idea of DA in the feature space, which is the foundation of our method, is not new as well. Nevertheless, our contribution is two-fold: (a) we propose the *first post-training* DA approach based on generative models that does DA *iteratively* in *difficult regions* of the learned representations to improve the generalization of deep networks. (b) we achieve *better results* than the state-of-the-art (SOTA) DA approaches on public benchmarks.

# 3  Methods

Let $D = \{\mathbf{x}_i, y_i\}_{i=1}^N$ be a labelled training dataset where $\mathbf{x}_i \in \mathbb{R}^d$ is the $i-th$ sample and $y_i \in \{1, \cdots, C\}$ is its associated class label over $C$ classes. Let $F_\theta : \mathbf{x}_i \to y_i$ be a deep network, parameterized by $\theta$, trained on $D$ with respect to a supervised loss, such as cross entropy, and an optimization procedure, such as stochastic gradient descent. Furthermore, let $E_{F_\theta} \in \mathbb{R}$ be the classification error of $F_\theta$ on a test set. The goal of this work is to implement an augmentation method $A$ to generate new training samples to fine-tune $F_\theta$ such that $E_{F_\theta} \geq E_{F_{\theta A}}$, where $E_{F_{\theta A}}$ is error of the model that is initially trained on $D$ and then fine-tuned using the augmented data generated by $A$.

## 3.1  Proposed Data Augmentation Method

Let us consider $F_\theta$ as a composition of two components. The first component is the parameterized representation learning function $H_\phi^L : \mathbf{x}_i \to f_i$, having $L$ representation learning steps or levels (e.g. convolutional layers or residual blocks), parameterized by $\phi$, which transforms input $\mathbf{x}_i$ to its corresponding feature representation (or embedding) $\mathbf{f}_i$. The second component is a linear or non-linear predictor function $P_\varphi^M : \mathbf{f}_i \to y_i$, having $M$ layers, parameterized by $\varphi$, which takes $\mathbf{f}_i$ as input and maps it to its corresponding class label $y_i$. Thus, $\theta = \{\phi, \varphi\}$, and $F_\theta$ is trained with respect to a supervised loss on $D$, optimizing the parameters of both components.

The proposed augmentation method is based on three key observations. Firstly, it is known that deep networks have the ability to exploit the hidden input to learn/discover meaningful representations at multiple levels such that the higher level features are defined in terms of low-level feature [21]. Secondly, it has been suggested that doing DA in the learned feature space, instead of the input space, can be more effective in improving the performance of supervised learning algorithms [7, 8]. Thirdly, learning difficulty associated with different instances in the feature space could be different [22].

Given these key observations, we propose the following DA method. We start by training $F_\theta$ on $D$ with respect to a supervised loss. Once trained, the DA is performed at multiple levels (in both low-level and high-level feature spaces) of $H_\phi^L$. The augmentation process is iterative, that is, the augmentations performed at level $l \in L$ are used to fine-tune the subsequent layers in both $H$ and $P$, before doing the DA at level $l + 1$.

## 3.2  Data Augmentation at Step $l$

Let $h^l(X)$ denote the learned representation or deep features (or simply activations) at level $l$ of $H_\phi^L$, where $X = \{\mathbf{x}_i\}_{i=1}^N$ is the set of training samples. We formulate the task of DA as the generation of new samples in $h^l(X)$. To achieve that, we set out an objective to learn a transformation function $g^l$ that transforms $h^l(X)$ into a compact space. We propose to learn a continuous space by modeling each vector in $h^l(X)$ as a distribution. Hence, we use variational inference to learn a variational family approximated through a variational autoencoder (VAE) [31], defined by

$$\hat{h}^l(\mathbf{x}) = g^l\left(h^l(\mathbf{x})\right), \mathbf{x} \in X, \tag{1}$$

where $g^l$ has two parts: the encoder $Q\left(\mathbf{z} \mid h^l\left(\mathbf{x}\right)\right)$ and the decoder $\bar{Q}\left(h^l\left(\mathbf{x}\right) \mid \mathbf{z}\right)$; $\mathbf{z}$ is the latent variable. We omit the argument $\mathbf{x}$ from $h^l$ for brevity, and define the loss function as

$$\mathcal{L}_{g^l} = \left\|\hat{h}^l - h^l\right\|^2 + KL\left(Q\left(\mathbf{z} \mid h^l\right) \| \mathcal{N}\left(0, I\right)\right), \tag{2}$$

where the first term is the reconstruction loss and $KL$ is the Kullback-Leibler divergence to measure the distance between the prior and the learned distribution. We assume that our proposed distribution, $Q\left(\mathbf{z} \mid h^l\right)$, is distributed as a Gaussian and try to minimize its distance towards a zero-centered unitary Gaussian, $\mathcal{N}\left(0, I\right)$). Finally, our objective is to learn a set of parameters for $g^l$ that minimizes Eq. 2, which is learned through backpropogation during the training phase of the VAE.

Once trained, we can use the encoder of the VAE to construct a set of distributions for the step $l$ as

$$\mathcal{Q}^l = \left\{(\mu, \sigma) = Q\left(\mathbf{z} \mid h^l\left(\mathbf{x}\right)\right) \mid \mathbf{x} \in X\right\}. \tag{3}$$

For a given $(\mu, \sigma)$, we can generate a new sample for augmentation by sampling a latent vector as

$$\mathbf{z} = \mu + \beta \sigma \epsilon, \epsilon \sim \mathcal{N}\left(0, I\right), \tag{4}$$

where $\epsilon$ is the generated noise, and we use $\beta$, which is randomly chosen from $[0, 1]$, to introduce variation in the synthesized sample. Each generated sample has a different $\beta$. Note that we tried both with and without $\beta$, and empirically found the former to work better. We believe that $\beta$ provides more powerful semantic transformations in the learned representations. The latent vector, thus sampled, is passed to the decoder to generate the augmented data point.

Though new data can be generated from every $(\mu, \sigma) \in \mathcal{Q}^l$, based on our third key observation (listed in section 3.1), we propose to generate new data for $y_i \in \{1, \cdots, C\}$ in regions where $y_i$ is hard to classify. In other words, we formulate the problem of generating new data from a point $\mathbf{p} \in h^l\left(X\right)$ if we believe that $\mathbf{p}$ is hard to classify. To achieve this, for every $\mathbf{p} \in h^l\left(X\right)$, we define a neighborhood as

$$\mathcal{N}_{\mathbf{p}} = \left\{\mathbf{q} \in h^l\left(X\right) \mid dist\left(\mathbf{p}, \mathbf{q}\right) \leq w\right\}, \tag{5}$$

where $dist\left(\mathbf{p}, \mathbf{q}\right)$ is the distance function for points $\mathbf{p}$ and $\mathbf{q}$. In this work, we implement it using the cosine similarity (CS) function. Note that we tried the Euclidean and Manhattan distances, too, but CS's results (reported in section 4.3) were slightly better.

Next, we compute the potential of $\mathbf{p}$ using radial basis functions (RBFs), as shown in Algorithm 1. The value of RBF of $\mathbf{p}$ and its neighbor $\mathbf{q}$ is added if they belong to different classes, otherwise it is subtracted. Therefore, observing a positive potential in $\mathbf{p}$ would mean that it is surrounded more by the instances of other classes as compared to the instances of the same class. Accordingly, we choose $\mathbf{p}$ as a source for generating new data if its potential is positive. Note that Algorithm 1 makes sure that at least one of its neighbors belongs to the same class as $\mathbf{p}$. This constraint is imposed to exclude potentially noisy samples. Thus, the final set of distributions for DA at step $l$ of $H_\phi^L$ is given as

$$\mathcal{O}^l = \left\{(\mu, \sigma) = Q\left(\mathbf{z} \mid h^l\left(\mathbf{x}\right)\right) \mid \mathbf{x} \in X, potential\left(h^l\left(\mathbf{x}\right) > 0\right)\right\}. \tag{6}$$

Finally, we generate the set of augmented data at step $l$ as

$$h_A^l = \left\{(\bar{\mathbf{p}}, y) \mid \bar{\mathbf{p}} = \bar{Q}\left(\mathbf{z}\right), \mathbf{z} = \mu + \beta \sigma \epsilon, (\mu, \sigma) \in \mathcal{O}^l, \epsilon \sim \mathcal{N}\left(0, I\right)\right\}, \tag{7}$$

where $y$ is the class label associated with $h^l\left(x\right)$ whose corresponding distribution is $(\mu, \sigma) \in \mathcal{O}^l$. Note that for every $(\mu, \sigma) \in \mathcal{O}^l$ we randomly generate $N_{(\mu, \sigma)}$ new data points (e.g., 3, 5 or 10 data points).

Once generated, we can use the $h_A^l$ in two ways. The first option is to freeze the representation learning component $H_\phi^L$, pass $h_A^l$ through $\left\{h^{l+1}, h^{l+2}, \cdots, h^L\right\}$ and use the output of $h^L$ to fine-tune only the predictor $P_\varphi^M$. The second option is to fine-tune all subsequent layers to optimize parameters

**Algorithm 1:** The algorithm to compute potential of a point $\mathbf{p} \in h^l(X)$

**Input** : Point $\mathbf{p}$, neighborhood $\mathcal{N}_{\mathbf{p}}$, and the spread of RBF $\gamma$

1   potential = 0
2   noisy = *true*
3   **for** *every* $\mathbf{q} \in \mathcal{N}_{\mathbf{p}}$ **do**
4     **if** $\mathbf{p}$ *and* $\mathbf{q}$ *have the same class label* **then**
5       noisy = *false*
6       potential = potential - $e^{\left(\frac{\|\mathbf{q}-\mathbf{p}\|}{\gamma}\right)^2}$
7     **else**
8       potential = potential + $e^{\left(\frac{\|\mathbf{q}-\mathbf{p}\|}{\gamma}\right)^2}$
9     **end**
10   **end**
11   **if** *noisy* **then**
12    potential = -1
13   **end**

**Output** : potential

of both the predictor and representation learning components. We tested both options, and found the latter to work significantly better.

After computing the $h_A^l$ and completing the fine-tuning of the proceeding levels, we repeat the entire process for level $l+1$. We present the pseudo-code of the proposed iterative hierarchical data augmentation procedure in Algorithm 2.

**Algorithm 2:** The IHDA Algorithm

**Input** : Dataset $D$, distance $w$, and probability $p$

1   Train $F_\theta = \left[H_\phi^L, P_\varphi^M\right]$ on $D$
2   **for** $l = 1$ *to* $L$ **do**
3    continue to next level with probability $\frac{1}{1+log(l)}p$
4    Train $g^l$ using Eq. 2
5    Compute $\mathcal{O}^l$ using Eq. 6
6    Generate $h_A^l$ using Eq. 7
7    Fine-tune $\left[\left\{h^{l+1}, h^{l+2}, \cdots, h^L\right\}, P_\varphi^M\right]$ using $h_A^l$
8   **end**

**Output** : $F_{\theta A}$

Note that the generic form of the proposed IHDA, as sown in Algorithm 2, can be applied to any deep network that has $L$ representation learning levels, where $l \in L$ can be a convolutional layer or even a residual block. In this work we tested IHDA mostly for block-based deep networks, i.e., residual networks, where we apply IHDA at block-level (output of residual blocks) instead of individual convolutional layers, except for one experiment (see sec. 4.2.2). Further note that to reduce overfitting and computational complexity for very deep networks, we employ an approach inspired by the idea of "dropout." More specifically, we drop out a level from IHDA with probability $p$, whose value can be found using a validation set. However, since feature spaces at deeper levels can be more meaningful than those at the earlier levels, we multiply $p$ with $\frac{1}{1+log(l)}$ to gradually reduce the probability of dropout for deeper levels.

## 4   Experiments

This section presents the results of the empirical validation of IHDA on three image classification benchmarks: CIFAR-10, CIFAR-100 [1], and ImageNet [32]. The CIFAR datasets consist of 32x32 colored natural images. CIFAR-10 has ten classes, where CIFAR-100 has undred classes. Both datasets have 50,000 images for training a model and 10,000 images for testing. On the other hand,

ImageNet has 1000 classes. It consists of 1.2 million images for training a model, and 50,000 images for testing.

This section also presents the results of applying IHDA on Sussex-Huawei Locomotion-Transportation (SHL) challeneg dataset [33], which is a (non-image) wearable-sensor based activity recognition benchmark. It consists of labelled data on eight different modes of transportation, which were collected from one person over a period of 82 days, from which 62 days of data is provided for training, and 20 days of data for testing. Both training and testing data have 1-minute segments ordered in a randomized fashion.

## 4.1 Experimental Setup

In the first experiment, we evaluated the performance of IHDA on CIFAR and ImageNet datasets with several state-of-the-art deep networks. For this, we implemented ResNet, Wide-ResNet, SE-RestNet, Shake-Shake, and PyramidNet. Within this experiment, we also compared the IHDA with the state-of-the-art DA methods on these datasets. For this, we chose AA [17], AAA [29], PBA [16], and ISDA [20]. The results of this experiment are summarized in Tables 1, 2 and 3.

In the second experiment, we conducted an ablation study to better understand the performance of different components of the IHDA on CIFAR datasets for RestNet-110. The results of this experiment are summarized in Table 4. Note that in all image-data experiments we evaluated both the standard IHDA and the IHDA+, where "+" indicates that the IHDA was applied with standard DA techniques. That is, we applied random compositions of the given set of transformation operations (translation, rotation, and flipping) to $D$ before training $F_\theta$ at step 1 of Algorithm 2. For a fair comparison, in IHDA+, the DA was performed at the same levels as those of IHDA.

Furthermore, to evaluate IHDA's performance on SHL dataset, we implemented a deep neural network following the guidelines given in [33], and tested it with and without IHDA. The input to the network were the frequency-domain features extracted from the raw magnitude vectors.

## 4.2 Implementation

This sections provides implementation details, including the details of hyper-parameter tuning for both image and SHL datasets.

### 4.2.1 Experiments on Image Data

In all experiments, the spread of RBF $\gamma$ in Algorithm 1 was set to 0.05. For other hyper-parameters (including $p$, and $w$), we held out a part of the training dataset as the validation set to find their optimum values. The hyper-parameter $p$ and $w$, for each experiment, are selected from the interval $[0, 1]$, with a step size of 0.05, based on the performance on the validation set.

For CIFAR datasets, the validation set had 5000 images, which were taken from the training set. For ImageNet, we used its reduced subset, which was created by randomly choosing 150 classes and 50,000 samples. From this reduced subset, we held out 5000 images for the validation set to tune the hyperparameters. The optimum values of the hyperparameters were then used to apply Algorithm 2 on full datasets from scratch. Note that the final results on ImageNet are reported on a set that is different from the validation set, which was used to tune the hyperparameters. For configurations of the models, please see the supplementary materials.

Three important points to note about the implementation of $g^l$ (a VAE at level $l$) are as follows. (1) For each $g^l$, we used the same architecture and hyper-parameters for all models, but it varied for CIFAR and ImageNet datasets. Please see the supplementary materials for details. (2) Each $g^l$ was trained on the output of residual block corresponding to level $l$. (3) In each experiment, to train $g^l$ on the output space of a residual block $l$, 10% of the set $h^l(X)$ was used as the validation set.

### 4.2.2 Experiments on SHL Data

In order to reproduce the results of [33], we followed their setup and considered only three sensors: accelerometer, gyroscope, and magnetometer. For each sensor, 1-minute multi-channel data were combined by computing a magnitude vector. Next, the magnitude vector for each 1-minute segment was truncated into 5-second frames with a skip size of 2.5 seconds. This resulted in 375,130 frames

Table 1: Test set error (%) of IHDA on CIFAR 10 with different models. Lower is better. We conducted five independent experiments, and report the mean values along with their standard deviations. The best results are **bold-faced**; whereas the second best results are *italic-faced*. For ISDA, PBA, AA and AAA, where available, we report the results from [20, 16, 17, 29], respectively. "Baseline (B)" represents the initial accuracy of IHDA+.

| Model | B | ISDA | PBA | AA | AAA | IHDA | IHDA+ |
|---|---|---|---|---|---|---|---|
| ResNet-32 | 7.60 | 7.09 | - | 4.50 | - | *3.90 ± 0.1* | **3.80 ± 0.1** |
| ResNet-56 | 6.67 | - | - | 3.60 | - | *2.93 ± 0.1* | **2.85 ± 0.1** |
| ResNet-110 | 6.45 | 6.33 | - | - | - | *2.78 ± 0.2* | **2.72 ± 0.2** |
| Se-ResNet-110 | 6.22 | 5.96 | - | - | - | *2.67 ± 0.2* | **2.50 ± 0.2** |
| Wide-ResNet-16-8 | 4.24 | 4.04 | - | - | - | *2.37 ± 0.1* | **2.23 ± 0.1** |
| Wide-ResNet-28-10 | 3.85 | 3.58 | 2.58 | 2.60 | **1.90** | 2.19 ± 0.1 | *2.11 ± 0.1* |
| Shake-Shake (26, 2x32d) | 3.55 | - | 2.54 | 2.50 | 2.36 | *2.07 ± 0.2* | **2.00 ± 0.2** |
| Shake-Shake (26, 2x96d) | 2.91 | - | 2.03 | 2.00 | 1.85 | *1.72 ± 0.2* | **1.66 ± 0.2** |
| Shake-Shake (26, 2x112d) | 2.84 | - | 2.03 | 1.90 | 1.78 | *1.63 ± 0.2* | **1.60 ± 0.2** |
| PyramidNet | 2.65 | - | 1.46 | 1.50 | 1.36 | *1.23 ± 0.1* | **1.20 ± 0.1** |

of training and 68,376 frames of testing data. Finally, Fast Fourier Transform was applied to each frame to extract a 753-dimensional feature vector.

For the classification model, we implemented a fully connected neural network classifier by following the guidelines provided in [33]. Since the model consisted of only three hidden layers, to which the IHDA was applied, we did not use the hyper-parameter $p$. The best hyper-parameter $w$ was selected in the same manner as it was done in experiments on image datasets using 10% of the training data as validation data. The spread of RBF $\gamma$ in Algorithm 1 was again set to 0.05.

Finally, each $g^l$ in this case also had the same architecture and hyper-parameters. Please see the supplementary materials for details. Similar to our experiments on image datasets, 10% of the set $h^l(X)$ was used as the validation set to train each $g^l$. However, unlike the experiments on image data, each $g^l$ in this case was trained on the activation space of a hidden layer, instead of a residual block, corresponding to level $l$.

## 4.3 Results

Table 1 presents the baseline results of the state-of-the-art deep networks on CIFAR 10. It also shows the results of state-of-the-art DA methods and those of IHDA and IHDA+. Please observe that the IHDA and the IHDA+ consistently improved the models' performance by iteratively fine-tuning them on the generated data. Furthermore, their performance is better than the existing DA solutions for these networks in all cases, except for Wide-ResNet-28-10 for which AAA gave the best results.

Between IHDA and IHDA+, the latter provided slightly better performance, which shows that the proposed IHDA algorithm benefited from doing simple augmentations in the input space. Therefore, we think that if used with more advanced DA methods, the IHDA can improve the generalization performance of deep networks even more. We confirm this fact by implementing IHDA+ using DA policies of AA [17], results for which are presented in section 4.4.

Table 2 and Table 3 present the results of IHDA and IHDA+ alongside the baseline performance of the deep networks on CIFAR 100 and ImageNet datasets, respectively. Similar to the results on CIFAR 10, these tables also show consistent improvement in the models' performances when fine-tuned with IHDA and IHDA+. Their results are also better than the state-of-the-art DA approaches, and the gain in the performance of IHDA with simple augmentations in the input space can be observed again.

Table 4 presents the ablation study results, which we did to understand the effectiveness of different components of IHDA. For this study, we considered the following variants: (1) $p = 0$ means that we did the augmentations for each $l$. (2) $p = 1$ means that we set the dropout probability to maximum. (3) *Only $P_\varphi^M$* means that we only fine-tuned the predictor part of the network. (4) *Same $p$ for each $l$* means that we did not reduce the dropout probability for deeper layers. (5) *Random Selection* means that instead of selecting source data points based on their potential, we selected them randomly.

Table 2: Test set error (%) of IHDA on CIFAR 100 with different models. The details are the same as those of Table 1. Note that the results of Shake-Shake are for its (26, 2x96d) implementation.

| Model | B | ISDA | PBA | AA | AAA | IHDA | IHDA+ |
|---|---|---|---|---|---|---|---|
| ResNet-32 | 31.30 | 30.27 | - | - | - | *22.20 ± 0.1* | **21.80 ± 0.1** |
| ResNet-110 | 28.57 | 27.57 | - | - | - | *18.45 ± 0.2* | **18.00 ± 0.2** |
| Se-ResNet-110 | 27.4 | 26.63 | - | - | - | *17.54 ± 0.2* | **17.00 ± 0.2** |
| Wide-ResNet-16-8 | 20.23 | 19.91 | - | - | - | *16.15 ± 0.1* | **16.00 ± 0.1** |
| Wide-ResNet-28-10 | 18.76 | 17.98 | 16.73 | 17.1 | 15.49 | *13.91 ± 0.1* | **13.19 ± 0.1** |
| Shake-Shake | 17.09 | - | 15.31 | 14.3 | 14.10 | *11.17 ± 0.2* | **11.08 ± 0.2** |
| PyramidNet | 14.01 | - | 10.94 | 10.70 | 10.42 | *09.70 ± 0.1* | **09.01 ± 0.1** |

Table 3: Validation set Top 1/ Top 5 accuracy (%) of IHDA on ImageNet with different models. Higher is better. We conducted three independent experiments, and report the mean values. The best results are **bold-faced**; whereas the second best results are *italic-faced*. Results for baseline and AA are taken from [17] and those of AAA are taken from [29].

| Model | Baseline | AA | AAA | IHDA | IHDA+ |
|---|---|---|---|---|---|
| ResNet-50 | 76.3 / 93.1 | 77.6 / 93.8 | **79.94** / 94.47 | 79.2 / 95.1 | *79.9 / **95.9*** |
| ResNet-200 | 78.5 / 94.2 | 80.0 / 95.0 | 81.32 / 95.3 | *82.0 / 96.8* | **82.9 / 97.3** |

Please observe that when augmented data were generated for each $l$, the performance of the model degrades, which could be because fine-tuning for each $l$ could lead to model's overfitting. The results of training only the predictor, using the same dropout probability for each level, and those of random selection of samples are better than the baseline, but not as notable as that of IHDA and IHDA+. This indicates the importance of tuning of all subsequent levels, forcing the dropout to decrease for higher levels, and generation of new data in difficult regions of the feature space, respectively. Note that the results of even the most light-weight version of IHDA ($p = 1$) were better than the baseline.

In the experiment on (non-image) SHL dataset, the baseline recognition accuracy of the model on the testing set was 83% (for before post-processing case), which is slightly better than what is reported in [33]. However, after fine-tuning the model with the IHDA, the performance improved to 92% ± 0.05%. Note that we report the mean performance of five independent trials. The results indicate that IHDA can be used to improve the generalization performance of deep networks for different domains.

### 4.4 IHDA+ using Augmentation Policies of AA [17]

We also tested IHDA+ with the learned augmentation policies of AA for (a) Wide-ResNet-28-10 on CIFAR-10 and (b) Resnet-50 on ImageNet. In (a) the test error (%) improved to 1.92 (previously 2.11). In (b) the Top 1/ Top 5 accuracy (%) improved to 81.47 / 96.50 (previously 79.9 / 95.9). These results confirm that if used with advanced DA methods, the IHDA can improve the generalization

Table 4: Test set error(%) of the ablation study of IHDA on CIFAR datasets for ResNet-110

| Setting | CIFAR 10 | CIFAR 100 |
|---|---|---|
| Baseline | 6.45 | 28.57 |
| $p = 0$ | 6.87 | 29.68 |
| $p = 1$ | 4.14 | 24.25 |
| Only $P_\varphi^M$ | 6.01 | 27.78 |
| Same $p = 0.55$ for each $l$ | 5.01 | 25.65 |
| Random selection | 5.73 | 26.91 |
| IHDA ($p = 0.55$) | *2.78* | *18.45* |
| IHDA+ ($p = 0.55$) | **2.72** | **18.00** |

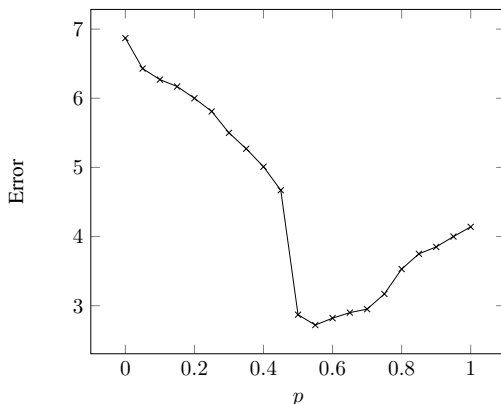

Figure 1: Test error (%) of ResNet-110 vs. $p$ on CIFAR-10

performance of deep networks even more. Therefore, IHDA can be considered as a complementary approach to the SOTA DA techniques that work in the input space

## 5   Hyper-parameters $p$ and $w$

For the hyper-parameter $p$, we observed a U-shaped behavior; see Figure 1. That is, increasing its values improved the performance to a certain extent, after which the performance started to degrade. Furthermore, we observed that the best value of $p$ depends on the number of levels $L$. For example, for ResNet-32, the best results were obtained for $p = 0.35$, whereas it was $p = 0.55$ for ResNet-110.

As for $w$, it is a difficult hyper-parameter to tune and has to be tuned for each $l$, separately. Unfortunately, it is not easy to provide a concrete guideline on how to select its optimal value other than using the validation set.

## 6   Computational Complexity

IHDA is an iterative method, which starts after the initial training of the model to convergence, where each iteration is a composition of *(a) Generation of augmented data* and *(b) Fine-tuning of the model*. However, the number of iterations is determined by the hyperparameter $p$, which can be tuned based on practical user constraints. Furthermore, each iteration fine-tunes a smaller version of the model (only proceeding layers are trained) on fewer data points (only points with positive potential are employed) as compared to the initial training. On average, computed over all experiments, IHDA took about 30% of the original training time, which also includes the time spent on tuning hyperparameters. For the sake of comparison, we trained the baseline model for ResNet-110 (without IHDA) for the same extra number of epochs on CIFAR-10 & CIFAR-100. The test errors were 6.33 and 28.21, respectively, which are significantly larger than those of IHDA and IHDA+.

## 7   Conclusion

In this paper, we proposed a domain-agnostic post-training domain augmentation (DA) method, called IHDA, to fine-tune trained deep networks to improve their generalization performance. In contrast to previous approaches, the IHDA performs iterative data augmentation in both low-level and high-level learned representation spaces of a deep network. At a given level, DA is achieved by transforming the feature space into a distribution space and generating new samples using the learned distributions. For effective DA, the IHDA synthesizes new samples in hard-to-learn regions by analyzing each data point's neighborhood properties. The new representations, thus generated, are used to fine-tune the parameters of the subsequent layers. Superior results on three image classification datasets and one (non-image) activity classification dataset demonstrate the effectiveness of IHDA in improving the generalization performance of deep networks.

## Broader Impact

In this paper, we proposed a new data augmentation technique to improve the generalization of any deep network, making our work general enough to be applied to a large variety of supervised learning problems. Since we do not foresee any particular application for our method, a Broader Impact discussion is not applicable.

## Acknowledgments and Disclosure of Funding

This research was funded by abcData.org. They also provided the necessary programming support.

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
