[Supplementary Material]

# Supplementary Materials

## A  Model Hyperparameters for Image Data Experiments

**CIFAR**  For ResNet and SE-ResNet, we used the same hyper-parameters as reported in [1]. However, for Wide-ResNet, Shake Shake and PyramidNet, we used the same hyper-parameters as reported in [2].

**ImageNet**  We tested two models: ResNet-50 and ResNet-200. For both of them, we used the same hyper-parameters as reported in  [3].

When fine-tuning a layer in all models, with IHDA or IHDA+, the initially used learning rate at that layer was divided by 10.

## B  Model and Its Hyperparameters for SHL Data Experiment

Configuration of the fully-connected neural network was taken from [4]. It consisted of a 753 dimensional input layer, followed by three hidden layers. Each hidden layer was a dense layer having 512 nodes, a batch normalization (BN) layer, a non-linear Relu layer, and a dropout layer that applied dropout with a dropout ratio of 25%. Finally, there was the decision layer that consisted of a dense layer of 512 nodes, a nonlinear Softmax layer and a a classification layer, which finally inferred the transportation mode of the current frame.

## C  Architecture of $g$, a VAE

In all cases of CIFAR and ImageNet, the input to $g$ was of size W x H x C, which represents the size of the output of a residual block.

**CIFAR**  The encoder network consisted of three convolutional (conv) layers and two dense layers. The size of the conv layers were: (16 x 16 x 128), (8 x 8 x 256) and (4 x 4 x 512), with padding = same, and Relu as the activation function. Each conv layer was followed by BN and a max pooling layer of size (2 x 2). The final conv layer was connected to two parallel dense layers. Each dense layer had 512 units, a BN layer and used Relu as their activation function. Their outputs were the mean and the variance vectors, respectively. The decoder network consisted of a dense layer of size 512, with Relu activation. It was followed by three deconvolutional layers of sizes (8 x 8 x 512) and (16 x 16 x 256), and (W x H x C), respectively, with BN, same padding, upsampling of size (2 x 2) and Relu as the activation function. Adam optimizer was used, with a learning rate of $3e - 5$. We trained the model for 30 epochs with a batch size of 128.

**ImageNet**  The encoder network consisted of four conv layers. The size of the conv layers were: (16 x 16 x 128), (8 x 8 x 256), (4 x 4 x 512), and (2 x 2 x 1024) with padding = same, and Relu as the activation function. Each conv layer was followed by BN and a max pooling layer of size (2 x 2). The final conv layer was connected to dense layer of size 1024, which was connected to two parallel dense layers of size 512. Each dense layer was followed by a BN layer and used Relu as their activation function. The decoder network consisted of two dense layers of sizes 512 and 1024, respectively. These were followed by four deconvolutional layers with sizes (4 x 4 x 1024), (8 x 8 x

512), (16 x 16 x 256), and (W x H x C), respectively, with BN, same padding, upsampling of size (2 x 2) and Relu as the activation function. Adam optimizer was used, with a learning rate of $3e - 5$. We trained the model for 60 epochs with a batch size of 256.

**SHL**   We reshaped the activation vector of a hidden layer into a 3D tensor as an input to $g$. The encoder network consisted of two conv layers and two dense layers. The size of the conv layers were: (64 x 4 x 4), and (128 x 2 x 2), with same padding, and Relu was used as the activation function. Each conv layer was followed by BN layer. Each dense layer had 64 units and Relu activation. Their outputs were the mean and the varaince vectors, respectively. The decoder network consisted of dense layer of size 64, with Relu activation. It was followed by two deconvolutional layers with sizes (128 x 2 x 2) and (64 x 4 x 4), respectively, with same padding and Relu as the activation function. Each layer was followed by BN layer. Adam optimizer was used, with a learning rate of $3e - 5$. We trained the model for 30 epochs with a batch size of 128.