[Reviews · NeurIPS 2020]

Review 1

Summary and Contributions: The paper introduces IHDA, an algorithm that enables post-training fine-tuning of visual and non-visual models by carefully augmenting features / activations of data examples.

Strengths: The key strength of the algorithm is its ability to improve both visual and non-visual models, something that is traditionally hard. IHDA improves CNN model accuracy on CIFAR-10, CIFAR-100 and ImageNet by a large margin. The approach introduced by authors is novel and is of interest to the community.

Weaknesses: Authors mention that w is a difficult hyper-parameter that requires selection for each layer, but they haven't provided any sensitivity analysis of it, or the set of values they used to train models. Exploration of distance functions other than cosine similarity would potentially shed further light on this problem. Authors discussed training of autoencoder in details, but did not talk about details of fine-tuning the main network, or how many examples were selected in O set (step 5 in algorithm 2). Those extra details would help understand the paper better.

Correctness: Some measurements have error bounds, while others (e.g. Table 4) don't. It would be great to add error bounds to as many measurements as possible.

Clarity: The paper is well written and is easy to follow along. The only hiccup I had was on line 207, where hyper-parameters p and w are mentioned. While I remembered p, I could not remember w and had to scan paper multiple times. It would be great to give them few extra descriptive words that far from where they are defined.

Relation to Prior Work: Authors contrast their work with data augmentation algorithms that modify input data directly (image augmentations), expert designed vs meta-learned techniques, explicit vs implicit techniques. Notably, they quote work of Terrance et al [7] and Li et al [29]. In [7], authors use autoencoder and apply linear transformations to encoded features. I think it would be useful to expand on differences with [7] in the Related Work section.

Reproducibility: Yes

Additional Feedback: On line 90, [8] -> [7]. ---------- Authors addressed most of my questions in the rebuttal. I'm raising my score to 7.


Review 2

Summary and Contributions: The authors propose an iterative hierarchical data augmentation (IHDA) to fine tune deep neural networks to improve their generalization performance.

Strengths: The area and problem this paper addresses is important and interesting - data augmentation has been shown as an effective and important subfield to improve the performance and generalization of NNs.

Weaknesses: - Lack of novelty. Augmenting data with the information extracted from hidden layers is not new (see Manifold Mixup). - Experimental results are not convincing. The comparisons are not done w/ the state-of-the-arts. - The three observations provided are somewhat trivial, which is not sufficient to motivate the proposed IHDA method.

Correctness: Yes.

Clarity: The paper is poorly written. - The proposed method is not clearly described. - No illustration is provided to help readers better digest the paper. - Symbols are poorly used (e.g., ‘training steps’ and the ‘depths/levels’ are two different concepts, and authors chose the same symbol to represent them. - Also, there are typos (e.g., line 6: hard-to-learn, not hard-to-lean) through the paper.

Relation to Prior Work: No. Several important previous works are missing - name few below: - mixup: Beyond empirical risk minimization - AutoAugment: Learning Augmentation Strategies from Data - Manifold Mixup: Better Representations by Interpolating Hidden States

Reproducibility: No

Additional Feedback:


Review 3

Summary and Contributions: This paper proposes a new data augmentation method that (1) performs data augmentation in feature space, (2) acts as a post-training procedure to fine-tune trained networks, and (3) fine-tuning the network in a layer-by-layer manner. Experiments show that the proposed method significantly outperforms AutoAugment on CIFAR-10, CIFAR-100, and ImageNet. On a wearable sensor-based activity recognition benchmark, the proposed method improves the baseline accuracy from 83% to 92%.

Strengths: - The proposed method at a high level looks reasonable: performing data augmentation in the feature space using VAE-based perturbation. - The experimental results on CIFAR-10, CIFAR-100, and ImageNet look significant and encouraging. - The experiments include an evaluation of a non-image dataset, demonstrating that the proposed method has the potential to work well across different domains.

Weaknesses: - The relationship between this work and the previous methods are not exposed. Since the idea of data augmentation in feature space is not new, I expect several papers closely related to this work. However, I cannot see which cited papers are closely related to this work and how it differs from them. - The computational complexity of the proposed method is not reported. (1) For example, how many times of fine-tuning is performed on average for one training job? How much relative time is required for the proposed fine-tuning comparing to the initial training? (2) Since the proposed method relies on cross-validation to determine the hyperparameters (p, w) for each level’s fine-tuning, it would be better to report the training time in detail, including the time spent on cross-validation. (3) Line 168 said, “ to reduce overfitting and computational complexity for very deep networks.” However, there is no comparison in terms of computational complexity. - The proposed method may be hard to reproduce. Since the proposed method is a multi-pass method, I feel that the sensitivity to hyperparameters could be somewhat higher than a single-pass method. For reproduction, it would be better if reporting the detailed hyperparameters (determined by cross-validation) in the supplementary material. For example, w used for each level l, and p used for each network architecture.

Correctness: The claims and the method look correct overall. Some minor concerns are discussed in the weakness question.

Clarity: This paper is well written and I enjoyed the reading.

Relation to Prior Work: The relation to prior work is somewhat unclear, which is one of the weaknesses mentioned above.

Reproducibility: No

Additional Feedback: How beta (in Equation 4) is sampled is not clearly expressed. In particular, for line 156 that says randomly generating N new data points, I do not know which of the following is performed: 1. Sampling a beta~U(0,1). Then draw N samples via eps~N(0, I). These N samples use the same beta. 2. Draw N samples via eps~N(0,I) and beta~U(0,1). Therefore, these N samples use a different beta. The necessity of beta is unclear. Line 137-138 said, “to introduce variation in the synthesized sample.” However, epsilon~N(0, I) also introduces variation. What is unique to beta? Why both beta and epsilon are needed? In the ablation study, for “Only P^M” and “Random selection,” what value of p is used? Typos: Line 6: hard-to-lean Line 104: f_i Equation 6: potential(h^l(x)>0) ======== post rebuttal ======== I read all the reviews and the rebuttal. In terms of my review, the authors addressed two out of the three weaknesses and all of the additional feedback. I would like to raise the score to 7 because of the significance of the new results provided in the rebuttal.


Review 4

Summary and Contributions: - The paper proposed a feature space data augmentation approach for further fine-tuning a trained network with the augmented feature representations. - A method to identify hard to learn regions for data augmentation. - Extensive experiments on CIFAR-10, CIFAR-100, and ImageNet, demonstrate the performance of the proposed approach.

Strengths: - Good performance on ImageNet; - Despite a bit complex, the method of identifying hard to learn regions is new.

Weaknesses: - When augmenting the latent features space, how to guarantee the semantics of the augmented representations are not changed? The author should explain and give more analysis about this. - The post training augmentation requires more training epoches when compared with the baseline approach and the compared SOTA. At least, the author is required to provide the results when the baseline model without augmentation is fine-tuned by the same epoches as the the post training process. - The paper should present comparisons with the recent published papers such as "ADVERSARIAL AUTOAUGMENT". The results presented in the paper are inferior to Adversarial autoaugment in several settings. The author should compare with this approch. - Hyperparameters in ImageNet. The hyper-parameters are tuned on the validation set. However, the results are only reported on the validation set. The hyperparameters have the potential to overfit to the validation set. I understand the ImageNet does not provide the test set. Could the author split some training data for validation? - The method seems to be very complicate. The author should report the fine-tuning time and compare the complexity with other compared approaches - Another big issue with this paper is the poor writing. A lot of typos and grammar errors such as: LN 6: Impelementing DA in hard to lean-> learn LN 111: meaningful representation -> representations LN 133: our objective is to learned .....

Correctness: Seems to be correct.

Clarity: No. See my comments in the weakness, Moreover the boarder Impact part is missed in this paper.

Relation to Prior Work: Yes

Reproducibility: No

Additional Feedback: See my weakness part.


Review 5

Summary and Contributions: This paper proposes an iterative fine-tuning method, where a separate VAE is trained to augment the representations of a trained network for each layer (level). The subsequent layers of the NN is fine-tuned based on the representation that is output from the VAE. This process is repeated for every layer in the representation. The paper achieves a significant improvement in generalization on CIFAR-10/100 and ImageNet.

Strengths: The improvements on the CIFAR-10/100 and ImageNet datasets are impressive. In previous work, generative models could not be utilized to improve the generalization of classification models (e.g. see Ravuri&Vinyals NEURIPS 2019). This paper achieves significantly higher generalization by utilizing a generative model, albeit the generative model is not trained on the input images but on the representations of a trained network. I think that this work could be of interest to NEURIPS community since: 1) it reports to achieve great results on ImageNet and 2) it shows an interesting approach to taking advantage of generative models.

Weaknesses: The method has no theoretical grounding, however this is not an issue for such a complex method. I'm glad that the authors did not try to make up irrelevant theoretical arguments for why their method might be working. 1) However, I worry that the method has some issues around empirical evaluation. First of all, I am not sure if the baselines considered in the paper are strong enough. This method trains a initial model to convergence, but then fine-tunes the model L steps, and a separate VAE is trained for each of those steps. The data augmentation methods they compare (especially the newer ones, see below) only train a model once. To me it seems like a more fair comparison would be a combination of good data augmentation and self-distillation, which also would re-train the model for several iterations. Pseudolabelling papers could also be a comparison, like Noisy-Student. (of course with the caveat that Noisy-Student uses extra data, and this paper does not). 2) I am also curious about why the authors did not compare to more recent and stronger data augmentation methods? (for example, Adversarial AutoAugment achieves better accuracies than PBA and AA, and RandAugment is cheaper). 3) I also find some of the important details missing. For example: a) What value of L was used for the reported models? b) What is the accuracy of their model before any of the fine-tuning steps, for IHDA and IHDA+? I assume the "Baseline" column might be the initial accuracy for IHDA, but this is not stated in the paper. What about the initial accuracy of IHDA+? c) The authors report that they could not evaluate better data augmentation methods for IHDA+ due to computational constraints, and instead used a combination of data augmentation operations which is not described well. Why not use a policy that is already opensourced by Fast AA, PBA, AA, or RA? I cannot imagine why training with these already opensourced policies would be any more expensive than what the paper currently uses for IHDA+. Furthermore, more details about IHDA+ would be appreciated, as well as its initial accuracy. d) This method is very expensive. That in itself is not a dealbreaker, but it would have been nice if the authors gave more information about how much more expensive this method is compared to only training a model once. I did not see any details reported on this.

Correctness: As mentioned above, some details are missing. But the reported details themselves do not suggest that any of the claims is incorrect. The empirical methodology sounds correct, but it would be great if the authors opensourced their code implementing the method.

Clarity: I think that the paper is well-written, except for the missing details that are listed above.

Relation to Prior Work: Of the listed previous work, the authors have clearly discussed how this work differs from them.

Reproducibility: Yes

Additional Feedback: Some minor suggestions: 1) It seems like the method in its most naive formulation, where P=0, does not actually improve generalization. However, with P=0.55 and P=1, the improvement is very large. I would love to see more data here. For example, can the authors plot final accuracy vs. P? 2) It again seems like selecting source data points based on their potential seems very important. It'd be great to see more details on this, as well as some more detail on how the potential is implemented during fine-tuning iterations. 3) There is a typo on line 6 of the abstract. 4) I'm not sure I understand the sentence that starts on line 25. "[The goal of data augmentation is] to minimize the distance between the training and the test sets." I don't think this is correct. Do the authors have references or results to support this claim? -------------------------------------------------------------------------------------- After author response period: I would like to thank the authors for posting their response to the reviews. The authors have provided more detail about their methodology, and showed that their results further improve if they use good augmentation strategies for the input images. For these reasons, I increased my score from 6 to 7. As mentioned originally, I was excited to see this paper as it did 2 things perhaps for the first time: 1) effective representation augmentation 2) effective utilization of a generative model for classification. I do hope that the authors will open-source their implementation, since there are lots of details to their method and it might be difficult for a reader to implement all of it correctly.

[Author Response · NeurIPS 2020]

**IHDA+ using Open-sourced Data Augmentation (DA) Policies:** We could not test IHDA with advanced DA methods as our initial experiments finished on the last day of the paper submission deadline, and we did not have extra resources to test this. Nevertheless, now we have tested IHDA+ with the learned augmentation policies of AutoAugment (AA) for (a) Wide-ResNet-28-10 on CIFAR (C10) and (b) Resnet-50 on ImageNet. In (a) the test error (%) improved to 1.92 (previously 2.11). In (b) the Top 1/ Top 5 accuracy (%) improved to 81.47 / 96.50 (previously 79.9 / 95.9). These results confirm that if used with advanced DA methods, the IHDA can improve the generalization performance of deep networks even more. Therefore, IHDA can be considered as a complementary approach to the SOTA DA techniques that work in the input space

**Comparison with Manifold Mixup (MM) and Adversarial Autoaugment (AAA):** We did not compare with MM as it was presented in the literature as a *regularization* technique. As for AAA, we thank the reviewer for pointing it out. Of all the experiments, AAA was better than IHDA & IHDA+ in just two cases (see previous comment). However, based on the new results, IHDA+ with DA policies of AA beats AAA in both settings. We will contrast IHDA with AAA and MM in our paper.

**Computational Complexity:** IHDA is an iterative method, which starts after the initial training of the model to convergence, where each iteration is a composition of *(a) Generation of augmented data* and *(b) Fine-tuning of the model*. However, the number of iterations is determined by the hyperparameter $p$, which can be tuned based on practical user constraints. Furthermore, each iteration fine-tunes a smaller version of the model (only proceeding layers are trained) on fewer data points (only points with positive potential are employed) as compared to the initial training. On average, computed over all experiments, IHDA took about 30% of the original training time, which also includes the time spent on tuning hyperparameters. For the sake of comparison, we trained the baseline model for ResNet-110 (without IHDA) for the same extra number of epochs on C10 & C100; the test errors were 6.33 and 28.21, respectively, which are significantly larger than those of IHDA and IHDA+.

**Error Plot vs. P:** Figure 1 presents test error (%) of ResNet-110 on C10 vs. $p$ for IHDA+.

**Novelty:** Neither the problem of DA is new, nor is the idea of DA in the feature space, which is the foundation of our method. Nevertheless, our contribution is two-fold: (a) we proposed the *first post-training* DA approach based on generative models that does DA *iteratively* in *difficult regions* of the learned representations to improve the generalization of deep networks. (b) we achieved *better results* than SOTA DA approaches on public benchmarks.

**Distance Function:** We tried cosine similarity (CS), Euclidean, and Manhattan distances. All gave similar results, but CS's results (reported in the paper) were slightly better.

Figure 1: Test error (%) of ResNet-110 vs. $p$ on C10

**Preserving Semantics of the Augmented Representations:** Although we might think that it is important to preserve the semantics of augmented representations , recent works [Ref:20 from the paper] have shown that DA provides better results if semantic transformations are allowed. In our work, we achieve this through $\beta$ and $\epsilon$ within a generative process.

**Combination of Good DA and Self-distillation for a Fair Comparison:** We agree that existing DA approaches train the model once; however, most of them do a fair amount of work before that. Nevertheless, we will certainly try to implement their advice and perform a comparison, but it would be extremely helpful if the reviewer explained their idea in more detail.

**How Many Examples were Selected in O:** As already mentioned on L. 156, we used every example in the set to generate new data points, since every example's potential is positive.

**Hyperparameters (HPs):** We will mention the values of all HPs in the supplimentary material as best as we can.

**Results on ImageNet:** The results on ImageNet are reported on a set that is different from the validation set, which was used to tune the hyperparameters. We will clarify this better in the paper.

**Initial Accuracy:** We have checked our implementation and found that the "Baseline" column represents the initial accuracy of IHDA+. We will also add to the paper the initial accuracy of IHDA.

**Beta:** Each generated sample has a different $\beta$. We tried both with and without $\beta$, and empirically found the former to work better. We believe that $\beta$ provides more powerful semantic transformations in the learned representations.

**Others:** In the ablation study, $P_\varphi^M$ and Random Selection both had $p = 0.55$. We will (a) include the error bounds for as many measurements as possible, (b) expand on differences with [7] in the Related Work, (c) get rid of all the typos.

[Meta-Review · NeurIPS 2020]

In this paper, the authors propose a new method for training deep networks that relies on learning a generative model for data augmentations. The resulting generating model is derived from a variational auto-encoder (VAE) and trained to learn augmentations for each hidden representation in the network. The authors test this new method for data augmentation and fine-tuning on CIFAR-10, CIFAR-100 and ImageNet. On all of these benchmarks, the authors identified substantial gains in terms of classification accuracy. The reviewers raised concerns in terms of the theoretical grounding of the method, the strength of the baselines, the training time in practice and clarity of presentation of the complex method. Some of these concerns were well addressed by the rebuttal. In significant post-rebuttal discussion, the reviewers concurred that providing a first demonstration of a generative model improving a classifier's performance on a challenging benchmark was impactful that may significantly affect many other forms of machine learning. This point subsequently convinced all reviewers to raise their scores accordingly. Although I do agree that the theoretical grounding of the method is limited and the method itself is quite complicated, the demonstration of this empirical result is an important contribution to the field and deserves broader recognition because it may spur others to investigate the benefits of generative models for data augmentation. That said, the authors have substantial work to perform in terms of incorporating results and discussion from the rebuttal as well as significantly improving the presentation and clarity of the manuscript. In this last point, I stress that the authors should pay close attention to address the suggestions of R2 to improve the clarity. Assuming that all of these issues are addressed, this paper is conditionally accepted to the NeurIPS conference.